# Invasive Aspergillosis among Lung Transplant Recipients during Time Periods with Universal and Targeted Antifungal Prophylaxis—A Nationwide Cohort Study

**DOI:** 10.3390/jof9111079

**Published:** 2023-11-04

**Authors:** Cornelia Geisler Crone, Signe Marie Wulff, Bruno Ledergerber, Jannik Helweg-Larsen, Pia Bredahl, Maiken Cavling Arendrup, Michael Perch, Marie Helleberg

**Affiliations:** 1Centre of Excellence for Health, Immunity and Infections (CHIP), Copenhagen University Hospital—Rigshospitalet, Blegdamsvej 9, 2100 Copenhagen O, Denmark; signe.marie.wulff.01@regionh.dk (S.M.W.); ledergb@gmail.com (B.L.); jannik.helweg-larsen@regionh.dk (J.H.-L.); marie.helleberg@regionh.dk (M.H.); 2Department of Infectious Diseases, Copenhagen University Hospital—Rigshospitalet, Blegdamsvej 9, 2100 Copenhagen O, Denmark; 3Department of Thoracic Anesthesia, Copenhagen University Hospital —Rigshospitalet, Blegdamsvej 9, 2100 Copenhagen O, Denmark; pia.bredahl.jensen@regionh.dk; 4Unit of Mycology, Statens Serum Institut, Artillerivej 5, 2300 Copenhagen, Denmark; maca@ssi.dk; 5Department of Clinical Microbiology, Copenhagen University Hospital —Rigshospitalet, Blegdamsvej 9, 2100 Copenhagen O, Denmark; 6Department of Clinical Medicine, University of Copenhagen, Blegdamsvej 3B, 2200 Copenhagen N, Denmark; michael.perch@regionh.dk; 7Department of Cardiology, Section for Lung Transplantation, Copenhagen University Hospital —Rigshospitalet, Blegdamsvej 9, 2100 Copenhagen O, Denmark

**Keywords:** aspergillosis, invasive aspergillosis, transplantation, lung transplantation, prophylaxis, triazoles, antifungal agents, voriconazole

## Abstract

The optimal prevention strategy for invasive aspergillosis (IA) in lung transplant recipients (LTXr) is unknown. In 2016, the Danish guidelines were changed from universal to targeted IA prophylaxis. Previously, we found higher rates of adverse events in the universal prophylaxis period. In a Danish nationwide study including LTXr, for 2010–2019, we compared IA rates in time periods with universal vs. targeted prophylaxis and during person-time with vs. person-time without antifungal prophylaxis. IA hazard rates were analyzed in multivariable Cox models with adjustment for time after LTX. Among 295 LTXr, antifungal prophylaxis was initiated in 183/193 and 6/102 during the universal and targeted period, respectively. During the universal period, 62% discontinued prophylaxis prematurely. The median time on prophylaxis was 37 days (IQR 11–84). IA was diagnosed in 27/193 (14%) vs. 15/102 (15%) LTXr in the universal vs. targeted period, with an adjusted hazard ratio (aHR) of 0.94 (95% CI 0.49–1.82). The aHR of IA during person-time with vs. person-time without antifungal prophylaxis was 0.36 (95% CI 0.12–1.02). No difference in IA was found during periods with universal vs. targeted prophylaxis. Prophylaxis was protective of IA when taken. Targeted prophylaxis may be preferred over universal due to comparable IA rates and lower rates of adverse events.

## 1. Introduction

Early survival following lung transplantation (LTX) is primarily challenged by graft rejection and infections [1,2]. Fungal infections in lung transplant recipients (LTXr) are predominantly caused by molds, with *Aspergillus* spp. being the primary pathogen [3,4,5,6]. Several circumstances in LTXr are believed to increase the risk of IA, including high immunosuppression and impaired mucociliary clearance in combination with the constant exposure of the transplanted organ to the ubiquitous *Aspergillus* spores through inhalation [3]. Invasive aspergillosis (IA) affects 4–15% of LTXr patients [7,8,9,10,11,12,13] and is most frequent in the first year after transplantation [9,10,12]. IA is associated with increased mortality [10,14]. There is no consensus on the best strategy for the prevention of fungal infections. A wide range of strategies for the prevention of IA after LTX are employed worldwide, including universal or targeted prophylaxis or pre-emptive therapy with different antifungal medications [15,16,17,18,19].

In many centers, triazoles are the preferred antifungal agents for prophylaxis with or without inhaled amphotericin B in different formulations [15,20]. Toxicity and drug–drug interactions are frequent during the use of triazoles and commonly result in the premature discontinuation of prophylaxis. For voriconazole, premature discontinuation rates range from 14 to 84% in LTXr cohorts [21,22,23,24,25,26]. Our center previously found that 62% of LTXr patients did not complete voriconazole prophylaxis per protocol [27]. While the newer triazoles, posaconazole and isavuconazole, have lower rates of toxicity and hold promise for prophylactic use [22,23], they are more expensive than voriconazole and drug–drug interactions remain a concern.

Despite several studies on antifungal prophylaxis in LTXr cohorts, there is insufficient evidence to confirm a protective effect of prophylaxis, as demonstrated in systematic reviews and meta-analyses [28,29,30]. 

A previous study from our center, which compared universal voriconazole prophylaxis to no prophylaxis in a study period from 2002 to 2006, did not show a preventive effect of prophylaxis [11]. Previous studies have compared the risk of IA between different prophylaxis protocols, regardless of the extent of premature discontinuation or duration of prophylaxis during follow-up. These “intention-to-treat” analyses add important information on the overall efficacy of an antifungal strategy. However, the question of whether prophylaxis is effective for individual patients while given is not answered. 

In our center, universal voriconazole prophylaxis was used until 2016, when a new guideline for targeted prophylaxis with posaconazole and inhaled liposomal amphotericin B was implemented. We aimed to compare the efficacy of these two prophylaxis guidelines in the prevention of IA overall and to evaluate the efficacy of antifungal prophylaxis during person-time with prophylaxis compared to person-time without prophylaxis in the two periods combined.

## 2. Materials and Methods

In this retrospective cohort study, we included all Danish patients, ≥16 years of age, receiving a lung transplantation in 2010–2019 in the Lung Transplantation Center, Copenhagen University Hospital Rigshospitalet, which is the only center performing LTX in Denmark. 

Data from nationwide registries were collected in the Centre of Excellence for Personalized Medicine of Infectious Complications in Immune Deficiency (PERSIMUNE) Data Warehouse [31] and consisted of data regarding transplantation and results on pathological, microbiological, and radiological examinations performed as part of clinical practice. Data on the prescription and discontinuation of antifungal prophylaxis, symptoms, and bronchoscopy findings were retrieved through a review of medical records and organized in a RedCap database [32].

### 2.1. Invasive Aspergillosis

Invasive aspergillosis (IA) was defined according to the ISHLT criteria [33] and classified as proven or probable pneumonia, tracheobronchitis, or anastomosis infection. In brief, the classification depends on the presence of positive microbiological/pathological findings of mold plus radiological or bronchoscopy findings and symptoms indicating *Aspergillus* infection. All patients with a record in the national pathological registry with a culture or histopathological finding of invasive mold in a tissue biopsy were classified as having proven IA.

All patients with a positive test for mold in the national microbiology registry were evaluated and classified according to the ISHLT criteria for IA in an expert panel review process, which is illustrated in Figure 1. 

#### Review Process for Classification of Invasive Aspergillosis

Cases of positive mold findings available in the national registries were entered for expert reviewing. The following was defined as a positive mold finding: one positive galactomannan antigen test in bronchoalveolar lavage fluid (BAL) of optical index ≥ 1 or blood of optical index ≥ 0.5; culture findings of either mold, *Aspergillus* species, or specified *Aspergillus* (one sample in BAL or two consecutive samples from the upper respiratory samples); a positive *Aspergillus* PCR in BAL. Cases with a positive finding were initially reviewed by two reviewers (CGC and SMW), where additional clinical findings in the medical record were evaluated and patients were grouped as potentially having IA or as being colonized. Cases with potential IA underwent further review, where two members of the expert review board classified the cases independently through a review of medical records. If the two reviewers reached an identical classification, this classification was final. If the two reviewers disagreed, the case was reviewed in plenary by the entire expert review board and discussed until a consensus on a final classification was reached. The expert review board consisted of senior medical doctors with expertise in the field (JHL, PB, MCA, MP, MH).

### 2.2. Prophylaxis and Immunosuppressive Protocols 

During the study period, the guidelines for antifungal prophylaxis changed: (1) from the start of the study period to July 2016, a universal voriconazole tablet of 200 mg twice a day was recommended for all patients during the first three months after transplantation; (2) from July 2016 to the end of the study period, the recommendation was targeted prophylaxis for patients at high risk of IA with a posaconazole extended-release tablet of 300 mg once a day and liposomal amphotericin B 25 mg inhalation during the first three months after transplantation. Patient groups categorized as having a high risk of IA included those with cystic fibrosis, sarcoidosis, retransplantation, and others (see Appendix B). Therapeutic drug monitoring (TDM) was not performed routinely during prophylaxis in either period. Pre-emptive treatment was not used systematically during the study period. A positive mold finding was handled at the clinicians’ discretion. 

Immunosuppressive therapy consisted of induction therapy with thymoglobulin and methylprednisolone and maintenance therapy with a calcineurin inhibitor (CNI), prednisolone, and an antimetabolite. Throughout the study period, cyclosporine was the primary CNI used, but 57 patients were randomized to receive either cyclosporine or tacrolimus during the ScanCLAD study from 2017 to 2019 [34]. The primary choice of antimetabolite was azathioprine (2010–2016) and mycophenolate (2016–2019).

Antiviral prophylaxis was prescribed as previously described [35]. Additional descriptions of the antimicrobial prophylaxis protocols are available in Appendix C. 

### 2.3. Routine Sampling and Microbiological Analyses

Bronchoscopy with bronchoalveolar lavage (BAL) and transbronchial biopsies were performed routinely at weeks two, four, six, and 12 and months six, 12, 18, and 24 after transplantation throughout the study period. Additional bronchoscopies were performed if clinically indicated. 

The bronchoalveolar lavage procedure followed international guidelines [36] and involved administering two 50 mL aliquots of sterile isotonic saline in the middle lobe or lingula, unless otherwise directed by abnormal imaging or airway exam, with the bronchoscope tip wedged in a segmental or sub-segmental airway. Aspiration was done immediately after the installation of saline and the return aliquots were pooled and submitted for clinical testing. All BAL samples were routinely sent for microbiological examination by microscopy and culturing (Sabouraud and blood plates). While specific examinations for fungi, such as galactomannan antigen tests and *Aspergillus* PCR (available from 2017), were not routine examinations, they were performed on clinical indication. Transbronchial biopsies and BAL samples were routinely sent for histopathological and cytological examination with Grocott-Gomori’s Methenamine Silver staining and microcopy.

### 2.4. Statistics

For the comparison of the distribution of variables in the universal and targeted prophylaxis periods, Fisher’s exact and Mann–Whitney U tests were used for categorical and continuous variables, respectively. Cases of proven and probable pneumonia, tracheobronchitis, and anastomosis infections were pooled into the combined outcome IA. Patients were followed from the date of LTX until IA, death/retransplantation, the end of the study period (31 December 2020), or one year after transplantation, whichever came first. The cumulative hazard of IA was calculated using the Nelson Aalen estimator. The risk of IA was analyzed in two multivariable Cox proportional hazard models. Variables included in the models were chosen prior to analyses based on existing evidence on factors associated with the outcome. The following variables were included in both models: sex, age > 50 years, *Aspergillus* pre-transplantation, high risk of IA (cystic fibrosis, sarcoidosis, or retransplantation), single lung transplantation, and calendar period (2010–July 2016 and July 2016–2019). In the second model, person-time with prophylaxis was included as a time-updated variable in addition to the other variables. The person-time with prophylaxis was started on the date of prescription of prophylaxis and ended 14 days after discontinuation. If a patient received antifungal treatment in relation to colonization, this was not included as person-time with prophylaxis.

One patient died on the first day after transplantation and was not included in the Cox models.

Sensitivity analyses were performed using the composite outcome of the first coming event of *Aspergillus* colonization or IA instead of IA only. In these analyses, two patients had colonization at the time of transplantation and were not included in the Cox models. 

We used Stata/SE 17.0 (StataCorp, College Station, TX, USA) and R version 3.6.1 for analyses.

The study was approved by the Danish Ethical Committee (H-20071557), the Danish National Board of Health (3-3013-1060/1), and the Danish Data Protection Agency (RH-2016-47).

## 3. Results

### 3.1. Baseline Characteristics

During the study period, 295 patients received an LTX, of whom 193 (65%) were transplanted during the universal prophylaxis period and 102 (35%) during the targeted prophylaxis period. Patient characteristics are summarized in Table 1. The median age was lower in the patients transplanted during the universal prophylaxis period (52 years, IQR 42–57) compared to the targeted period (55 years, IQR 45–58). There were more patients with cystic fibrosis as the underlying disease in the universal prophylaxis period, N = 36 (19%), compared to the targeted prophylaxis period, N = 8 (8%).

During the universal prophylaxis period, 183 (95%) patients initiated antifungal prophylaxis, whereas only six (6%) patients initiated prophylaxis in the targeted prophylaxis period. Of the patients starting voriconazole prophylaxis in the universal prophylaxis period, 38% completed ≥9 of the intended 12 weeks of prophylaxis. The median duration of prophylaxis was 37 days (IQR 11–84) and 108 days (IQR 44–142) during the universal and targeted prophylaxis periods, respectively. The time on prophylaxis was 16% and 3% of the total follow-up time, in the universal and targeted prophylaxis period, respectively (Figure 2).

### 3.2. Invasive Aspergillosis

In the first year following transplantation, IA was diagnosed in 27 (14%) and 15 (15%) patients, during the universal and targeted prophylaxis period, respectively. Of these, IA was classified as proven in 41% and 47%, respectively. The distribution of IA as per ISLHT category is summarized in Appendix A and Figure 3. The majority of IA cases were caused by *Aspergillus fumigatus* (Appendix A).

The median time to IA was 101 days (IQR 47–172) during the universal prophylaxis period and 84 days (IQR 14–101) during the targeted prophylaxis period (*p* = 0.11). There were three cases of breakthrough infection during prophylaxis, two in the universal prophylaxis period (both *A. fumigatus*) and one in the targeted prophylaxis period (unspecified mold). Resistance to the prophylactic triazole was not found in any of these breakthrough infections. 

IA manifested as pneumonia in 15 and 7 patients during the universal and targeted prophylaxis period, respectively. The median time to IA pneumonia was 101 days (IQR 36–160) vs. 14 days (IQR 12–15) in the universal vs. targeted period and 7/15 (47%) vs. 3/7 (43%) had died by the end of follow-up (1 year after LTX).

Colonization after LTX was detected in 27 (14%) of the patients in the universal period and 14 (14%) in the targeted period (*p* = 0.86). Two and one of the colonized patients progressed to IA in the universal and targeted period, respectively. Antifungal treatment was started in 14 (52%) and 6 (40%), *p* = 0.53, of the colonized cases in the universal and targeted period, respectively. 

### 3.3. Associations between Prophylaxis and Invasive Aspergillosis

The cumulative hazards of IA in the two prophylaxis periods are visualized in Figure 4. There was no evidence of a difference between the two periods, with an adjusted hazard ratio (aHR) of 0.94 (95% CI 0.49–1.82) (Table 2).

In the second model including person-time on prophylaxis, being on prophylaxis was associated with a lower risk of IA, although it did not reach statistical significance, with aHR 0.36 (95% CI 0.12–1.02) (Table 2).

The results of sensitivity analyses with the combined outcome of colonization and IA showed similar trends (Appendix A).

## 4. Discussion

In this nationwide study of a large LTXr cohort, we compared the rates of IA within one year after LTX in time periods with universal vs. targeted antifungal prophylaxis. The person-time during which patients received prophylaxis was shorter than expected in both periods due to high rates of premature discontinuation in the universal prophylaxis period and low adherence to guidelines for the start of prophylaxis in the targeted prophylaxis period. We found a high incidence of IA, with no difference in rates between the prophylaxis periods, but there was a clear trend towards lower rates of IA during person-time on prophylactic treatment compared to time without prophylaxis.

The optimal antifungal prophylaxis strategy remains a discussion point in the field of LTX, where several different regimes are used worldwide [15]. The three major types of preventive strategies are universal prophylaxis, targeted prophylaxis, and pre-emptive therapy [19]. We compared the efficacy of a strategy of universal vs. targeted antifungal prophylaxis and found comparable rates of IA in the two periods. In the universal prophylaxis period, more patients had cystic fibrosis and received a single LTX, which are known risk factors of IA. These differences could lead to a higher a priori risk of IA in the universal period, and a protective effect of universal prophylaxis could thereby be masked when comparing the two periods one to one. However, there was no difference in the rates of IA after adjustment for these risk factors. 

Colonization is also known to be associated with IA, but we did not see a significant difference in cases of colonization during the two periods, and the number of colonizations leading to antifungal treatment, in a pre-emptive manner, was also similar. Additionally, sensitivity analyses did not indicate that the misclassification of IA vs. colonization or pre-emptive treatment influenced the results.

A few smaller previous studies have compared the efficacy of preventive strategies, with inconsistent results [37,38,39]. Koo et al. compared universal prophylaxis with inhaled amphotericin b (N = 82) to pre-emptive therapy with a systemic antifungal drug based on positive mycological findings early after transplantation (N = 83) [37]. The authors found a reduction in fungal infections in the peri-transplant period with pre-emptive therapy compared to universal prophylaxis. In a study by Linder et al. including 105 LTXr, a universal prophylaxis strategy of itraconazole +/− inhaled amphotericin B was compared to targeted prophylaxis with either voriconazole or fluconazole/micafungin based on the risk of mold [39]. During the 18-month follow-up period, the risk of invasive fungal infections (IFD) was significantly lower with universal prophylaxis compared to targeted prophylaxis, but the differences did not reach statistical significance in the sub-analyses with IA as the only outcome. Another recent study by Ju et al. compared universal voriconazole vs. posaconazole prophylaxis for the prevention of breakthrough IFD in 182 LTXr cases [40]. TDM was used for dose adjustment. There was no difference in breakthrough IFD between the two groups (9/142 vs. 1/40, *p* = 0.35). 

In the present study, we did not find evidence of a difference in protective effect between the two prophylaxis strategies. However, it is important to note that our study revealed low rates of utilization of prophylaxis in both prophylaxis periods, which should be taken into consideration when interpreting the results. The high rates of premature discontinuation of voriconazole were mainly due to hepatotoxicity, as previously discussed [27]. In Denmark, all medication related to transplantation is subsidized through the public health care system; thus, financial cost is not a barrier preventing patients from completing prophylaxis. Only a small proportion of the patients in the targeted period received prophylaxis (N = 6). Several patients were not started on prophylaxis even though they qualified as being at high risk of IA, according to the targeted prophylaxis guidelines. The targeted guideline was created by a national medical council and includes a broad variety of IA high-risk criteria, such as renal impairment, CMV infection, and antilymphocyte therapy (see Appendix A). Strict adherence to the guidelines would therefore, in practice, have resulted in nearly all patients being considered as high-risk patients. The implementation of a targeted guideline in our center was primarily motivated by a need to reduce the overall use of antifungal agents. During the preceding universal prophylaxis strategy period, side effects were frequent and an evaluation of voriconazole prophylaxis vs. no prophylaxis had shown no effect of prophylaxis [11]. These motivations likely led to an additional selection and individual risk assessment by the treating physicians, after the targeted guidelines were implemented. This discrepancy between guidelines and practice points to the need for the improved, accurate identification of patients at high risk when a targeted prophylaxis strategy is sought.

In our previous evaluation of adverse events related to toxicity and drug–drug interactions, we found that universal prophylaxis was associated with an increased risk of side effects and acute rejections [27]. When evaluating the clinical drawbacks and benefits of antifungal prophylaxis, adverse events should be taken into consideration, as well as efficacy, for a balanced assessment.

The overall cumulative incidence of IA in this study was approximately 15%. This is a rather high incidence compared to previous reports of IA in LTXr ranging from 4 to 15%, but most frequently below 10% [5,7,8,9,10,11,12,13,22,41,42]. Our findings may reflect the actual higher incidence compared to other LTX centers, perhaps due to the limited use of antifungal prophylaxis, or it could be related to the near-complete identification of IA events in our cohort. We identified all patients with positive mycological findings through a systematic, electronic search in the nationwide microbiology and pathology databases, which may have led to a higher degree of identification of IA events than in previous studies relying on the active reporting of events by study investigators, or where there was no access to complete nationwide data. The proportion of proven IA in the present study was approximately 40%, which exceeds previous studies, including reports from a large multicenter study of 900 LTXr where 19% of IA events were proven [10]. This supports the validity of the high incidence of IA found in our center, rather than this being related to a potential tendency towards classification as probable IA events vs. colonization. The poor protective effect of the two prophylaxis regimes, resulting in high incidence in both periods, may be related to several factors, including the poor utilization of prophylaxis, lack of TDM, environmental predisposition, or a large proportion of patients with underlying high-risk disease such as cystic fibrosis. 

To further elucidate why universal prophylaxis was not associated with lower rates of IA, we asked the following question: are rates of IA lower while LTXr are on prophylactic antifungal treatment compared to time without prophylaxis? Our analyses showed a protective effect during person-time with prophylaxis, indicating that high rates of the premature discontinuation of prophylaxis may be an important factor in explaining the observed limited effect of universal voriconazole prophylaxis.

TDM during triazole prophylaxis was not used in our center. This may have led to subtherapeutic levels compromising the protective effect. TDM is not routinely used to monitor prophylaxis in many LTXr centers, despite being recommended [15,16,43]. In a worldwide survey of LTXr centers from 2011, 26% used TDM [15], and in a survey of LTXr centers in the U.S. from 2019, 50% reported the use of TDM [16]. Some studies have shown a large proportion of subtherapeutic levels of triazoles in patients with breakthrough infections [44], and attention should be paid to the emergence of azole resistance [45] when the antifungal prophylaxis strategy is chosen. In the present study, there were only a few breakthrough infections and antifungal resistance of the prophylactic drug used was not found in any of these cases. The lack of TDM in our center could also potentially have contributed to the high rates of premature discontinuation of prophylaxis due to the failure in identifying supratherapeutic levels of triazoles leading to toxicity. A clinical trial randomizing patients treated with voriconazole for IA to TDM or no TDM found equal numbers of adverse events in the two groups, but a significantly lower rate of discontinuation in the group where TDM was performed [46].

The results of our study reflect some of the challenges of the currently available antifungal drugs, including poor tolerability and variations in metabolization. Antifungal agents with a profile better suited for prophylaxis are needed. New antifungal drugs are being developed and tested, including promising candidates like the first-in-class olorofim [47,48], the novel triazole opelconazole optimized for inhalation [45], and amphotericin B delivered by alternative methods [49].

Strengths of the study include the rather large cohort with complete nationwide electronic data on microbiological and pathological analyses, as well as a standardized screening protocol that remained unchanged during the study period. A systematized and thorough review process by clinical experts was used for the classification of IA. The study also had some limitations. It was retrospective, and the classification of IA relied on reports in medical records of clinical findings and diagnostic workup. The results of the analyses comparing two different prophylaxis strategies in different time periods may have been confounded by other factors that changed over time, such as advances in surgical techniques or other areas of transplant medicine that could affect the risk of IA, as previously reported by Peghin et al., demonstrating a decrease in the incidence of IA over time without changes in prophylaxis protocol [12]. This could have led to an underestimation of the protective effect of universal prophylaxis relative to the later period with targeted prophylaxis. Our center participated in a randomized controlled trial from 2017 to 2019, during which 57 patients were randomized to two different immunosuppressive regimes, which could have affected the risk of infection. Although the study had a larger sample size than most previous studies, the number of outcomes was relatively small, which limits the statistical power. Further investigations are needed in larger multicenter studies, preferably in a randomized controlled trial, to determine the optimal strategy for the prevention of IA.

## 5. Conclusions

We found no difference in the rates of IA among LTXr during periods with universal vs. targeted antifungal prophylaxis strategies, which might be explained by the high rates of premature discontinuation of prophylaxis due to toxicity. However, antifungal prophylaxis was protective of IA when taken. 

Targeted prophylaxis, for high-risk patients only, may be preferred over universal prophylaxis after LTX due to comparable rates of IA and lower rates of adverse events. 

## Figures and Tables

**Figure 1 jof-09-01079-f001:**
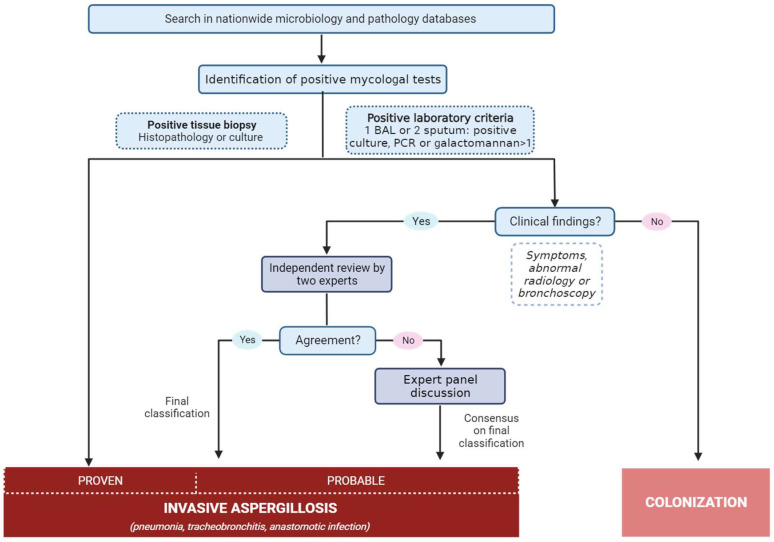
Method for classification of invasive aspergillosis based on ISHLT definition criteria.

**Figure 2 jof-09-01079-f002:**
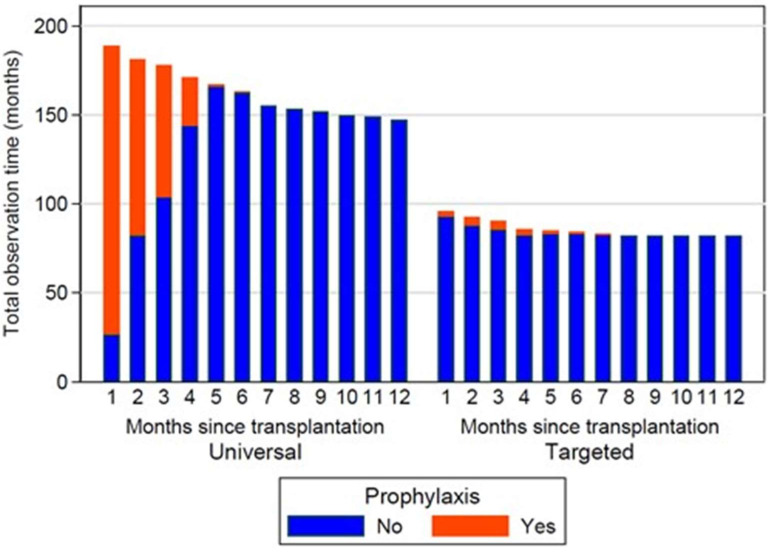
Person-time on and without antifungal prophylaxis during time periods with universal and targeted antifungal prophylaxis.

**Figure 3 jof-09-01079-f003:**
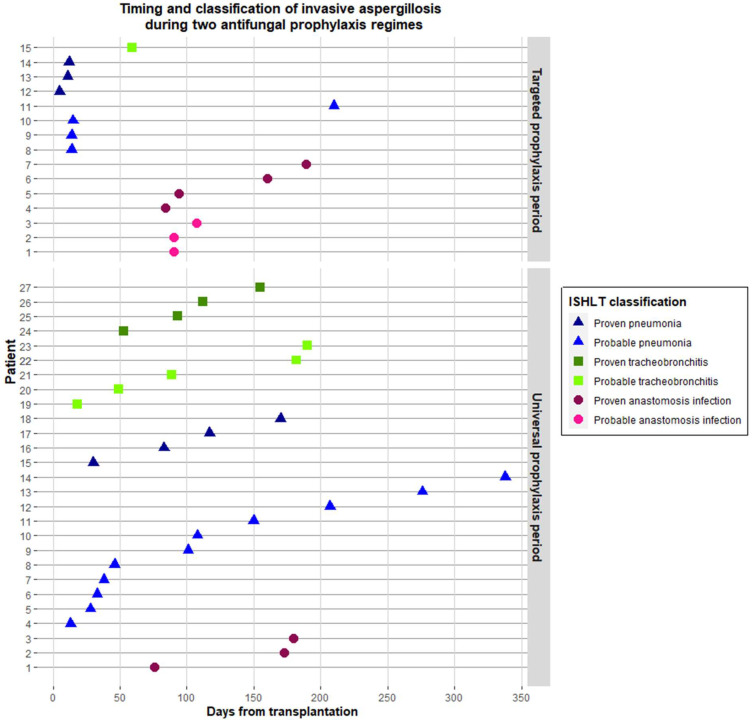
Timing and ISHLT classification of invasive aspergillosis during periods with universal and targeted antifungal prophylaxis guidelines.

**Figure 4 jof-09-01079-f004:**
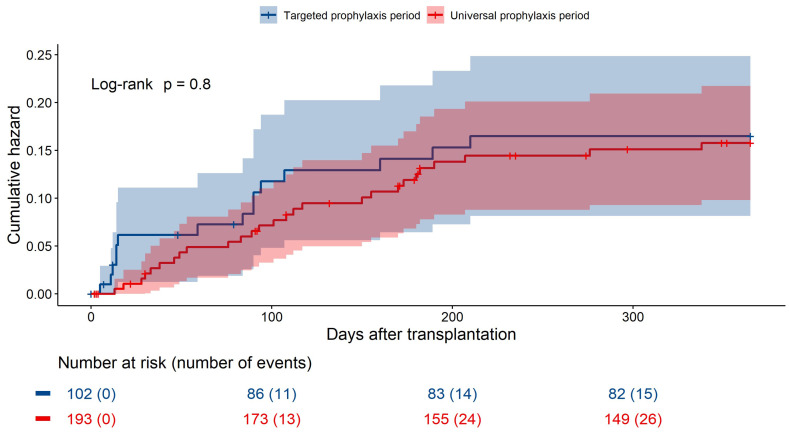
Cumulative hazards of invasive aspergillosis the first year after lung transplantation in time periods with universal versus targeted prophylaxis.

**Table 1 jof-09-01079-t001:** Characteristics of lung transplant recipients in the time periods with universal and targeted antifungal prophylaxis.

	Universal Prophylaxis Period, 2010–2016 (N = 193)	Targeted Prophylaxis Period, 2016–2019 (N = 102)	*P*-Value
**Male, *n* (%)**	106 (55)	46 (45)	0.11
**Age, years, *median* (IQR)**	52 (42, 57)	55 (45, 58)	0.04
**Underlying disease, *n* (%)**			
**Emphysema**	84 (44)	52 (51)	0.27
**Primary pulmonary hypertension**	5 (3)	5 (5)	0.32
**Pulmonary fibrosis**	50 (26)	33 (32)	0.28
**Cystic fibrosis ***	36 (19)	8 (8)	0.02
**Retransplantation ***	6 (3)	1 (1)	0.43
**Sarcoidosis ***	12 (6)	3 (3)	0.28
**Single lung transplant, *n* (%)**	25 (13)	5 (5)	0.04
***Aspergillus* prior to transplantation, *n* (%)**	18 (9)	6 (6)	0.49
**Initiated antifungal prophylaxis, *n* (%)**	183 (95)	6 (6)	<0.001
**Completed ≥ 9/12 weeks of prophylaxis** **, *n* (%)**	69/183 (38)	4/6 (67)	0.20

* Underlying diseases categorized as high risk of invasive aspergillosis (IA) qualifying for targeted prophylaxis. Invasive aspergillosis classified according to the ISHLT criteria, including pneumonia, tracheobronchitis, and anastomosis infections. Abbreviations: Universal antifungal prophylaxis period = voriconazole three months following transplantation for all patients; targeted antifungal prophylaxis = posaconazole and inhaled liposomal amphotericin B three months following transplantation for high IA risk patients.

**Table 2 jof-09-01079-t002:** Incidence rates and hazard ratios of invasive aspergillosis in lung transplant recipients.

Incidence Rate of IA per 100 Person Years of Follow-Up (95% CI)		Model 1 Universal vs. Targeted Period		Model 2 Person-Time on vs. without Prophylaxis	
Universal period	Targeted period	HR (95% CI)	aHR ^1^ (95% CI)	HR (95% CI)	aHR ^2^ (95% CI)
16.6 (11.4–24.2)	17.5 (10.7–29.1)	0.92 (0.49–1.73)	0.94 (0.49–1.82)	0.39 (0.15–1.01)	0.36 (0.12–1.02)

^1^ Model 1 adjusted for: sex, age > 50 years, *Aspergillus* pre-transplantation, high risk of IA, single lung transplantation, prophylaxis guideline period. ^2^ Model 2 adjusted for: sex, age > 50 years, *Aspergillus* pre-transplantation, high risk of IA, single lung transplantation, prophylaxis guideline period, person-time on prophylaxis.

## Data Availability

The data sets generated and analyzed during this study are derived from patients treated in Denmark. The data sets contain sensitive patient data governed by the General Data Protection Regulation and Danish law. Because of Danish legislation and approvals granted by the Danish Data Protection Agency, it is not possible to upload the raw data to a publicly available database. However, access to these data can be made available from the corresponding author on reasonable request, provided that a relevant data processing agreement is entered into according to current regulations.

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
