# Peer review of "Invasive Aspergillosis among Lung Transplant Recipients during Time Periods with Universal and Targeted Antifungal Prophylaxis—A Nationwide Cohort Study"

_jof, 2023, doi:10.3390/jof9111079_

Round 1
Reviewer 1 Report
Comments and Suggestions for Authors
Crone et al. performed a retrospective review to compare the efficacy of universal prophylaxis versus target prophylaxis strategies for the prevention of early aspergillosis in lung transplant recepients. They find no difference in rates of aspergillosis between the two strategies. Overall, this is a well-written study which adds to the data in order to help indiviudalize antifungal prophylaxis post-transplant. I especially commend the authors on their rigorous review of cases and subjecting review of cases to a review panel to provide the best definition of cases. This is not easy, and is often needed in mycosis research! I have some comments/questions that I think can improve the manuscript that I hope the authors will consider.
1) Page 2, line 14 reads "...regardless of extend of premature discontinuation...". This doesn't read right. Is it supposed to say extent or is this an extra word?
2) In Table 1, I would include a row of patients who were considered "high risk" to compare (in addition to the already included diagnoses).
3) If targeted prophylaxis was supposed to be given for all high risk patients post-2016, why were there only 6? Just from the high risk diagnoses there are 12. As well, one of the high risk factors in Appendix 1 is antilymphocyte treatment and in section 2.2 the authors state that "...induction therapy with thymoglobulin...". This would imply that all patients were high risk? Please clarify these points beyond what is already included in the discussion.
4) One may expect lower rates of tracheobronchitis and anastomotic infections with the use of inhaled amphotericin. Although there was a lower rate of tracheobronchitis in the targeted group, there seems to be higher rates of anastomotic infections. Why do you think that is?
Reviewer 2 Report
Comments and Suggestions for Authors
In the manuscript “Invasive aspergillosis among lung transplant recipients during time periods with universal and targeted antifungal prophylaxis – a nationwide cohort study,” the authors study the prevention of invasive aspergillosis in lung transplant recipients through the evaluation of the effectiveness of universal prophylaxis versus targeted prophylaxis. It is a well-planned study in which patients with invasive aspergillosis were selected through a systematized review by clinical experts, including complete data on microbiological and pathological analyses. The work is interesting since it is evident that antifungal prophylaxis protects against invasive aspergillosis when the treatment scheme is completed. However, I have some observations:
Discussing why invasive aspergillosis is a risk factor among lung transplant recipients would be interesting.
On the other hand, it is important to consider the increasing number of patients at risk of AI and a greater number of patients exposed to broad-spectrum azoles, such as voriconazole, posaconazole, or isavuconazole, through prophylaxis or as part of treatment in clinical practice, could be a factor contributing to a significant increase in azole resistance worldwide, which may lead to the limited effect observed with universal voriconazole prophylaxis. Furthermore, “cryptic” species are probably underestimated since conventional diagnostic methods often do not allow their identification; furthermore, the resistance mechanisms are still unknown for these species, and it has been observed that they present different susceptibilities to antifungals. Therefore, considering these aspects, they should expand their discussion.
Line 402-405: Uniform the style of the font.
References
Carefully review the references and standardize them according to the journal's format.
Reviewer 3 Report
Comments and Suggestions for Authors
A few minor points that may not be that important.
The authors state in the discussion that TDM is not always used to monitor prophylaxis in lung transplant centers, but the survey referenced is from 2011. Is there nothing more recent?
When the authors state that some studies have shown a high proportion of subtherapeutic levels of triazoles in patients with break-through infections, their reference is to the severe condition of vertebral osteomyelitis.
